# Lipid Profile Characterization and Lipoprotein Comparison of Extracellular Vesicles from Human Plasma and Serum

**DOI:** 10.3390/metabo9110259

**Published:** 2019-11-01

**Authors:** Yuchen Sun, Kosuke Saito, Yoshiro Saito

**Affiliations:** Division of Medical Safety Science, National Institute of Health Sciences, Kanagawa 210-9501, Japan; yuchen.s@nihs.go.jp (Y.S.); yoshiro@nihs.go.jp (Y.S.)

**Keywords:** extracellular vesicles, lipidomics, lipoproteins, mass spectrometry

## Abstract

Extracellular vesicles (EVs) consist of lipid bilayers, occur in various biofluids, and are invaluable in biomarker screening. Liquid chromatography coupled with high-resolution mass spectrometry (LC-MS) was recently used to study comprehensive EV lipid profiles in vitro. The aim of this study was to establish a lipidomics platform for human plasma and serum EVs for comprehensive characterization of their lipid profiles, and to compare them with those of other lipid-containing particles, such as high-density lipoproteins (HDL), and low/very low-density lipoproteins (LDL/VLDL). Isolation was validated by specific protein markers; CD9 and MHC class I for EVs, apoA-I for HDL, and apoB-100 for LDL/VLDL. Lipidomics identified 264 lipids from isolated plasma EVs, HDL, and LDL/VLDL. The absolute lipid levels per unit protein content in the EVs were more than eight times lower than those of the lipoproteins. Moreover, the EVs had higher lysoglycerophospholipid levels than HDL or LDL/VLDL. Similar profiles were also determined for human serum. The present study found that the lipid profiles of EVs are unique and distinctly different from those of lipoproteins. The lipidomics platform applied to human plasma and serum EVs could generate important information for the exploration and qualification of biomarkers in disease diagnosis.

## 1. Introduction

Lipidomics is a mass spectrometry-based technique that characterizes lipid profiles in biological samples. It has been used to screen for diagnostic biomarkers of metabolic conditions, neurological disorders, and cancers. Lipidomics has also been used to identify drug safety biomarkers, to elucidate pathological mechanisms, and for the discovery of novel therapeutic targets [1,2,3,4]. Whole plasma and serum are often used in lipidomics as they are easily and readily obtained with minimal invasiveness. However, they each contain different lipid carriers, such as extracellular vesicles (EVs) and lipoproteins. Alterations in the lipid profiles of the whole plasma and serum do not reflect the changes in specific lipid carriers. For this reason, detailed characterization of the lipids, and more specifically, the circulating lipid carriers, may be more suitable and effective for biomarker screening.

EVs are cell-derived lipid bilayer vesicles that participate in cell-to-cell communication under a pathophysiological state [5,6]. EVs are generated by multivesicular bodies and plasma membranes. EVs derived from the former are called exosomes (d = 40–150 nm). Those originating from the latter are known as microvesicles (d = 100–1000 nm) [7]. EVs contain various proteins, nucleic acids, and lipids. They have been used in biomarker screening [8,9]. The main research foci for EVs have been proteins, mRNAs, and miRNAs. Nevertheless, several types of lipids, such as cholesterol and glycosphingolipids, play important roles in exosome and microvesicle biogenesis [6,10,11,12,13,14]. Recent untargeted lipidomics reports disclosed that >200 lipids and 20 lipid classes could be detected in EVs isolated from cancer cells [15,16,17]. Another study detected 107 lipids in the urinary EVs from prostate cancer patients. The combination of phosphatidylserine (PS) (18:1/18:1), PS (18:0/18:2), and diglycosylceramide (CerG2) (d18:1/16:0), may serve as a diagnostic biomarker for prostate cancer [18]. Using untargeted and targeted lipidomics, Tao et al. found that several phospholipids in plasma EVs were associated with pancreatic cancer at the tumor stage and overall survival [19]. As with proteins and RNAs, the lipids in EVs are potential biomarkers for human diseases. Until now, however, most of the untargeted lipidomic studies on EVs were performed in vitro. Thus, our knowledge of the lipid profiles of circulating human EVs is very limited.

Lipoproteins are major lipid–protein complexes that transport lipids such as water-insoluble cholesterols and triglycerides (TG) [20]. Lipoproteins are categorized into several classes according to density, size, and apolipoprotein content. These include very low-density lipoproteins (VLDL), low-density lipoproteins (LDL), and high-density lipoproteins (HDL). LDL and VLDL transfer lipids from the liver to the peripheral tissues. HDLs participate in reverse cholesterol transport. According to the extracellular vesicle guidelines published in 2018 [21], no research has been conducted to compare the lipid profiles of EVs and lipoproteins in human plasma and serum. LDL and HDL may be copurified with the EVs collected by conventional ultracentrifugation-based isolation [22,23]. Therefore, separate characterization of the lipid profiles of circulating EVs and lipoproteins could elucidate the features of EVs in human plasma and serum.

Here, we have established a lipidomics platform for human plasma and serum EVs and isolated EVs without any apparent lipoprotein contamination. We also summarized the methodological differences of previous EV lipidomics used human plasma and/or serum [19,24], and compared them with our platform, as seen in Table 1. The results showed that EVs and lipoproteins each have their own unique lipid profiles. For example, lysoglycerophospholipids are abundant in EVs but not in lipoproteins. We also provide evidence that both plasma and serum can be assessed with our lipidomics platform. Thus, we present a new approach for the comprehensive study of the glycerophospholipid and sphingolipid profiles of the EVs, HDL, and LDL/VLDL isolated from human plasma and serum. This method could serve to screen for new lipid-based disease biomarkers.

## 2. Results

### 2.1. Validation of Isolation Method for EVs and Lipoproteins

To characterize the profiles of the blood lipid carriers, we isolated EVs from pooled human plasma and serum obtained from healthy donors. Transmission electron microscopy (TEM) and western blot showed that the isolated plasma and serum EVs were round and expressed the EV markers CD9 and MHC class I, but not the ER marker calnexin, as seen in Figure 1A,B. NTA confirmed the size distribution of the isolated EVs. The calculated modal and mean sizes of the plasma EVs were 97.8 ± 2.8 nm and 118.7 ± 2.3 nm while those of the serum EVs were 104.0 ± 2.9 nm and 117.2 ± 0.7 nm, respectively, as seen in Figure 1C. These data indicated that the EV fraction was successfully prepared from the plasma and serum.

After we isolated the EVs from the human plasma and serum, we then isolated the lipoproteins (HDL and LDL/VLDL) from the same sources, using the same biological matrices to compare them to the EVs. There was no detectable expression of either the apoB-100 (LDL/VLDL marker) or apoA-I (HDL marker) in the isolated EVs, as seen in Figure 1D. Thus, there was negligible lipoprotein contamination in the isolated EVs.

### 2.2. Comparison of EVs and Lipoproteins in Terms of Absolute Lipid Levels Per Unit Protein Content

We detected 311 and 264 lipids in the test sample (EVs and lipoproteins isolated from pooled plasma and serum) and the validation sample (EVs and lipoproteins isolated from the individual plasma and serum of the 12 healthy subjects), respectively, as seen in Table 2. In the test sample, we detected 206 glycerophospholipids and 105 sphingolipids. In the validation sample, there were 168 glycerophospholipids and 96 sphingolipids. Ten classes of glycerophospholipids were detected: lyso-PC (LPC), ether-type LPC (LPCe), lyso-PE (LPE), lyso-PI (LPI), PC, ether-type PC (PCe), oxidized PC (PC+O), PE, ether-type PE (PEe), and PI. Eleven classes of sphingolipids were identified: SM, oxidized SM (SM+O), Cer, oxidized Cer (Cer+O), deoxyCer (deoCer), glycosylCer (CerG1), oxidized CerG1 (CerG1+O), diglycosylCer (CerG2), triglycosylCer (CerG3), ganglioside (GM3), and sulfatide (ST). The top five most abundant individual lipids per class in the EVs are summarized in Appendix A. SM was the largest number in sphingolipid class and SM (d34:1; d18:1/16:0) was the most abundant SM. It constituted 26.88% and 26.46% of the SM class in the plasma and serum EVs, respectively. PC was the largest number in glycerophospholipid class and PC (34:2; 16:0/18:2) was the most abundant PC. It constituted 24.03% and 23.22% of the PC class in the plasma and serum EVs, respectively. More than 60 neutral lipids, including ChE, DG, and TG, were detected in our lipidomics platform (data not shown). In addition, total levels of cholesterolesters in the EVs were over 20 times lower than those in the lipoproteins, as seen in Appendix A, which support our experimental finding that there was negligible lipoprotein contamination in the isolated EVs. However, we excluded them from the statistical analyses as the relative SD for their corresponding internal standards (ChE (12:0) for ChE, DG (12:0/12:0) for DG, and TG (16:0/16:0/16:0-13C3 for TG)) were >30%.

To characterize the EV and lipoprotein lipid profiles, we performed PCA using the absolute lipid levels per unit protein content. The EVs, HDL, and LDL/VLDL were separated by the combination of components 1 and 2 in the test and validation samples, as seen in Figure 2A and Appendix A. Thus, these fractions differ in terms of their lipid profiles. Moreover, we found no separation between the plasma and serum EVs, HDL, or LDL/VLDL in the test or validation samples. We also compared the total lipid levels among the EVs, HDL, and LDL/VLDL. In the human plasma and serum, the EVs had the lowest lipid levels (more than eight times lower than those of the lipoproteins, as seen in Figure 2B and Appendix A. Therefore, the EVs are protein-enriched particles, whereas the lipoproteins are highly lipid-enriched particles. As the trends were similar between the test and validation samples, we used only the validation test in the subsequent statistical analyses.

### 2.3. Comparison of mol% Lipid Composition of EVs and Lipoproteins in Human Plasma and Serum

In addition to the absolute lipid levels, we also compared the mol% lipid compositions of the isolated EVs, HDL, and LDL/VLDL. PCA disclosed that the EVs, HDL, and LDL/VLDL were separated by the combination of components 1 and 2, as seen in Figure 3A. No separation was detected between the plasma and serum in terms of EVs, HDL, and LDL/VLDL. For this reason, there is no critical difference between the plasma and serum in terms of the lipid compositions of their EVs, HDL, and LDL/VLDL. Therefore, we focused mainly on the plasma in the subsequent experiments. Data acquired for the serum are summarized in Appendix A.

Having established distinct differences in mol% lipid composition between the EVs and the lipoproteins, we then examined the differences in their lipid classes. The levels of all sphingolipids except Cer+Os were lower in the plasma EVs than the plasma LDL/VLDL, as seen in Figure 3B. Relative to HDL, the levels of several sphingolipids, GM3s, and SMs, were significantly higher in the EVs. Collectively, the sphingolipid content was highest in the LDL/VLDL, followed by the EVs, and then the HDL. On the other hand, the EVs had higher lysoglycerophospholipid (LPCs, LPCes, and LPEs) content than LDL/VLVL and HDL, as seen in Figure 3C. Moreover, the levels of PCe, PE, PEe were lower in the EVs than the lipoproteins. PC was higher in the HDL than the EVs or LDL/VLDL. PC+Os was higher in the LDL/VLDL than in the HDL or the EVs.

We extracted the unique differences in the mol% individual lipid compositions among the EVs and the lipoproteins by drawing a heatmap with stringent statistical criteria, as seen in Figure 4. The measured differences in the individual lipid contents among the EVs and lipoproteins were almost consistent with the mol% lipid class compositions. The lipids accounting for the relatively higher PC composition in the HDL than the EVs or the LDL/VLDL contained polyunsaturated fatty acids (PUFA; 20:4, 20:5, and 22:6). These included PC (38:6a; 18:2/20:4, 18:1/20:5), PC (40:7b; 18:1/22:6), and PC (40:8; 20:4/20:4). The types and levels of these PCs in the LDL/VLDL were comparable to those in the EVs. In addition, the lipids accounting for the relatively higher PC composition in the LDL/VLDL or HDL compared with the EVs, were clearly different. The PCes that contained saturated- and monounsaturated fatty acids (SFA; 16:0 and MUFA; 18:1) were higher in LDL/VLDL, while those that contained PUFA, were higher in HDL.

### 2.4. Differences in Unsaturated LPC, PC, and PCe Content in the EVs and Lipoproteins

Lipid particle membrane fluidity varies with the degree of unsaturation. Thus, we classified the LPCs, PCs, and PCes into three groups based on the number of double bonds in their fatty acid side chains: SFA; no double bond, MUFA; one double bond, and PUFA; >1 double bond. We then calculated the MUFA:SFA and PUFA:SFA ratios. As shown in Figure 5A,B, the MUFA:SFA and PUFA:SFA of the LPCs and the MUFA:SFA of the PCs in the EVs were comparable to those in HDL and higher than those in LDL/VLDL. On the other hand, the PUFA:SFA of the PCs and the MUFA:SFA and PUFA:SFA of the PCes in the EVs were lower than those in HDL but higher than those in LDL/VLDL, as seen in Figure 5B,C. However, the PUFA:SFA of the PCs did not significantly differ between the EVs and HDL.

## 3. Discussion

In the present study, we have established a method to isolate EVs from human plasma and serum that enables a comprehensive lipidomics-based analysis of their glycerophospholipid and sphingolipid profiles. To the best of our knowledge, this is the first report comparing the lipid profiles of circulating EVs and lipoproteins. When we contrasted EVs with separately isolated HDL and LDL/VLDL, we found that: (a) the EVs isolated by the method developed in this study presented with negligible HDL and LDL/VLDL contamination; (b) the absolute lipid level per unit of protein content in the EVs was more than eight times lower than those of the lipoproteins; (c) the EV lipid profiles markedly differed from those of the lipoproteins; (d) a high lysoglycerophospholipid content is characteristic of circulating human EVs; and (e) comparable results were obtained for human plasma and serum.

When lipidomics are conducted on purified fractions, adjustments according to the protein content are usually done, to normalize input sample volumes. Our study demonstrated that EVs contain far less lipid than lipoproteins. Moreover, HDL and LDL/VLDL have substantially different lipid profiles from EVs. Thus, even moderate lipoprotein contamination results in levels of lipoprotein-derived lipids that could possibly surpass those in the EVs and simulates intrinsic lipid alterations in the EVs, following the adjustment of the input sample volume with the protein content. Therefore, lipoprotein contamination must be vigilantly monitored during EV lipidomics analysis. Both LDL and HDL could possibly be copurified with EVs isolated by conventional ultracentrifugation [22,23]. Thus, this method may not be suitable for lipidomics analyses of human plasma and serum EVs.

To the best of our knowledge, this is the first report demonstrating high lysoglycerophospholipid content in circulating human EVs. Nevertheless, several reports showed that EVs derived from cell cultures contained higher lysoglycerophospholipid levels than the parental cell lines [16,17]. Thus, there is a common mechanism accounting for the elevated lysoglycerophospholipid levels in EVs but it has not yet been elucidated. Phospholipase A2 (PLA2) generates lysoglycerophospholipids by breaking the sn-2 ester bonds of glycerophospholipids. An earlier proteomics study reported that PLA2s are loaded into EVs [25] and may generate lysoglycerophospholipids in EVs from glycerophospholipids.

PS is a component of cell cultures- and urinary-derived EVs and constitutes 5.4–14.9% of the total lipid content in them [7,16,17,18,19]. In the present study, however, no PS was detected in the EVs isolated from the healthy human plasma and serum. As we included an internal PS standard (PS (12:0/12:0)) in the lipid extraction and detected it at levels resembling those for the other internal standards, we ruled out the possibility that our lipidomics platform was the cause of the failure to detect PS. Using the same lipidomics platform, we detected PS in cell-based EVs following ultracentrifugation (data not shown). Therefore, the PS levels in the circulating EVs of healthy humans are probably too low to be detected by our lipidomics platform. A recent report recommended PS-positive EVs as markers for ovarian carcinoma as they were absent in normal patients [26]. It is also possible that our isolation method may have purified predominantly PS-negative EVs. PS-negative EVs were identified in healthy human plasma by flow cytometry and electron microscopy with PS-binding antibodies or proteins [27,28]. Nevertheless, none of these mechanisms could be determined using the PureExo exosome isolation kit. To clarify the presence of PS in the EVs from human plasma and serum, analysis using a different EV purification method is required, such as PS-binding receptor Tim4 and anti-EV marker antibody-based affinity isolation.

The lipid profiles of the plasma and serum HDL and LDL/VLDL were also addressed in the present study. HDL participates in reverse cholesterol transport. The lipid composition of the HDL observed here might be associated with its functional properties. Our results demonstrated that HDL had a higher mol% of PC than LDL/VLDL. This finding corroborates those previous reports [29,30]. We also showed that the PC PUFA:SFA ratio was higher in the HDL than the LDL/VLDL. It was stated earlier that the PC content in the recombinant HDL was positively correlated with the cellular cholesterol efflux rate [31,32]. Another study indicated that the degree of unsaturation in the PC acyl chain modulates the movement of free cholesterol from the cells to HDL [33,34]. It is, therefore, possible that higher PUFA-PC content promotes the efflux of cholesterol from the peripheral tissues to the HDL.

Our lipidomics platform for human plasma and serum EVs and lipoproteins detected and characterized >250 glycerophospholipids and sphingolipids. We also generated fundamental data on the characteristic lipid profiles of EVs and lipoproteins. Our approach may help identify the etiologies of various human diseases based on the lipid compositions of the circulating EVs that could serve as novel biomarkers. Our lipidomics methodology could also be applied towards lipoprotein research and help clarify the roles of these biomolecules in lipid-related metabolic diseases.

## 4. Materials and Methods

### 4.1. Human Plasma and Serum

Human plasma and serum were purchased from ProMedDx (Norton, MA, USA). Written and signed informed consent was obtained from all donors. Venous blood was collected in the morning after overnight fasting. Plasma samples were stored in vacutainer plasma separator tubes containing EDTA (Becton-Dickinson, Franklin Lakes, NJ, USA). Serum samples were stored in vacutainer plasma separator tubes containing clot activators (Becton-Dickinson, Franklin Lakes, NJ, USA). Plasma and serum were separated within 2 h of blood collection, immediately frozen, and stored at −80 °C. Plasma and serum samples were pooled as a test sample and divided into five replicates to validate the isolation and lipidomics methods. Individual human plasma and serum (validation) samples were used only in lipidomics analysis. Validation sample data are listed in Table 3. This study was approved by the Ethics Committees of the National Institute of Health Sciences and performed in accordance with the Declaration of Helsinki.

### 4.2. Cells and Cell Culture

HeLa cells were obtained from ATCC (Manassas, VA, USA) and cultured in RPMI 1640 medium (Wako Pure Chemical Industries Ltd., Osaka, Japan) supplemented with 10% (*v*/*v*) heat-inactivated fetal bovine serum (FBS) and antibiotics (100 U penicillin and 100 µg streptomycin). The cells were grown at 37 °C under a 5% CO_2_ atmosphere.

### 4.3. Extracellular Vesicle (EV) and Lipoprotein Isolation

Before EV isolation, plasma and serum were centrifuged at 3000× *g* and 4 °C for 15 min to remove cell debris. EVs were then isolated from 500 μL plasma and serum with a PureExo isolation kit (101Bio, Mountain View, CA, USA) according to the manufacturer’s protocol. The isolated EVs were resuspended in phosphate-buffered saline (PBS) and used in the subsequent assays.

For lipoprotein (HDL and LDL/VLDL) isolation, cell debris was removed from the plasma and serum by centrifugation at 10,000× *g* and 4 °C for 20 s. Lipoproteins were then isolated from 400 μL plasma and serum using the LDL/VLDL/HDL purification kit (Cell Biolabs, San Diego, CA, USA) according to the manufacturer’s protocol. The isolated HDL and LDL/VLDL were diluted with PBS and used in the subsequent assays.

Protein concentrations in the isolated EVs, HDL, and LDL/VLDL were determined by Bio-Rad protein assay (Bio-Rad Laboratories, Hercules, CA, USA). Approximately 80 μg of the EVs and 10 μg of the HDL or LDL/VLDL were subjected to lipid extraction.

### 4.4. Transmission Electron Microscopy

Transmission electron microscopy (TEM) was performed with the HITACHI H-7600 (Tokyo, Japan) at 100 kV to visualize the EVs isolated from the human plasma and serum. Isolated EV samples (~5 μL) were adsorbed onto a 400-mesh carbon-coated copper grid (NISSHIN EM, Tokyo, Japan) for 10 s and negatively stained with 2% (*w*/*v*) uranyl acetate for 10 s. Excess liquid was removed with absorbent paper. The grids were air-dried before TEM observation. The electron micrograph output was a 2D projection of the negatively stained specimens.

### 4.5. Western Blot

Lysate of whole HeLa cells was prepared with radioimmunoprecitation assay (RIPA) buffer (Wako Pure Chemical Industries Ltd., Osaka, Japan). The lysate was used in Western blot. Isolated EVs, HDL, and LDL/VLDL were diluted and heated to 95 °C in LDS sample buffer (Thermo Fisher Scientific, Waltham, MA) for 5 min. The protein content of the EV markers (CD9, MHC-class I) and the endoplasmic reticulum (ER) marker (calnexin) was 30 μg in each case. The protein content was 100 ng for the LDL (apoB-100) and 20 ng for the HDL (apoA-I) markers. The proteins were separated by sodium dodecyl sulfate-polyacrylamide gel electrophoresis (SDS-PAGE) using NuPAGE 4–12% Bis-Tris gel (Thermo Fisher Scientific, Waltham, MA, USA) and MES SDS running buffer (Thermo Fisher Scientific, Waltham, MA, USA). The separated proteins were transferred to a polyvinylidene fluoride (PVDF) membrane (GE Healthcare, Chicago, IL, USA) which was then blocked with 5% (*w*/*v*) bovine serum albumin (Wako Pure Chemical Industries Ltd., Osaka, Japan) in Tris-buffered saline with 0.2% Tween (5% (*w*/*v*) BSA/TBS-T) at room temperature for 1 h. The primary antibodies were mouse anti-CD9 IgG (200 × dilution; sc13118; Santa Cruz Biotechnology, Dallas, TX, USA), mouse anti-HLA class I (MHC class I) IgG (500 × dilution; ab70328; Abcam, Cambridge, UK), rabbit anti-calnexin IgG (1000 × dilution; 2433s; Cell Signaling Technology, Danvers, MA, USA), rabbit anti-apolipoproteion B-IgG (10,000 × dilution; ab139401; Abcam, Cambridge, UK), and rabbit anti-apolipoprotein A-I (10,000 × dilution; ab52945; Abcam, Cambridge, UK). The secondary antibodies were horseradish peroxidase (HRP)-conjugated goat anti-rabbit IgG (2000 × dilution; sc2030; Santa Cruz Biotechnology, Dallas, TX, USA), and HRP-conjugated goat anti-mouse IgG (2000 × dilution; sc2005; Santa Cruz Biotechnology, Dallas, TX, USA). All antibodies were diluted with 2.5% (*w*/*v*) BSA/TBS-T. Proteins were detected by chemiluminescence using ECL western blot detection reagents (GE Healthcare, Chicago, IL, USA) and ImageQuant LAS 4000 mini (GE Healthcare, Chicago, IL, USA).

### 4.6. Nanoparticle Tracking Analysis

Nanoparticle tracking analysis (NTA) was performed with Nanosight LM10 (Malvern Instruments, Amesbury, UK) to measure isolated EV particle sizes. Each sample was recorded 10 × for 1 min each time at camera level = 15 and detection threshold = 4. The data were analyzed in NTA v. 3.2 (Dev Build 3.2.16; Malvern Instruments, Amesbury, UK). Other parameters such as blur size and maximum jumping distance were set to automatic mode.

### 4.7. Lipid Extraction

Lipids were extracted in an automated system (Microlab NIMBUS; Hamilton Robotics, Reno, NV, USA). Fifty-microliter samples were mixed with 550 μL methanol/isopropanol (1:1) containing 2 μM of each of the following internal standards: phosphatidylcholine (PC) (12:0/12:0); phosphatidylethanolamine (PE) (12:0/12:0); phosphatidylinositol (PI) (15:0/18:1-d7); phosphatidylglycerol (PG) (12:0/12:0); PS (12:0/12:0), phosphatidic acid (PA) (12:0/12:0); cardiolipin (CL) (14:0/14:0/14:0/14:0); sphingomyelin (SM) (d18:1/12:0); ceramide (Cer) (d18:1/12:0); glycosylceramide (CerG1) (d18:1/12:0); sulfatide (ST) (d18:1/12:0); diacylglycerol (DG) (12:0/12:0) (Avanti Polar Lipids, Alabaster, AL, USA), cholesterolester (ChE) (12:0); and triacylglycerol (TG)(16:0/16:0/16:0-13C3) (Larodan Fine Chemicals, Solna, Sweden). The 400 μL sample mixtures were deproteinized with FastRemover Protein (GL Science, Tokyo, Japan). Then, 250 μL eluent was added to 500 μL water and 3% (*v*/*v*) formic acid. The resulting 500 μL mixtures were loaded into FastRemoverC18 (GL Science, Tokyo, Japan). The plates were washed with 400 μL water/methanol (1:1). The lipids were eluted with 500 μL methanol/isopropanol (1:1) followed by 500 μL methanol/isopropanol (1:1) with 3% (*v*/*v*) ammonia. Then, 800 μL eluent was evaporated and dissolved in 50 μL methanol for lipidomics analysis.

### 4.8. Lipidomics Analysis

Lipidomics analysis was performed according to a previously reported method [3] using a 10 μL injection volume. Briefly, extracted lipids were separated and analyzed using reverse phase liquid chromatography (RPLC: Ultimate 3000, Thermo Fisher Scientific) connected with high resolution MS (Orbitrap Fusion, Thermo Fisher Scientific). An InertSustainSwift C18 column (3 μm, 2.1 × 250 mm [P]; GL Science, Tokyo, Japan) was used, and temperature of the column oven was set to 55 °C. The mobile phase A and B was water/methanol/acetonitrile (21:20:60, *v*/*v*/*v*) supplemented with 0.1% formic acid and 10 mM ammonium formate, and water/methanol/acetonitrile (1:10:90, *v*/*v*/*v*) supplemented with 0.1% formic acid and 10 mM ammonium formate, respectively. The mobile phase was pumped through at a flow rate of 0.25 mL/min. The multi-step gradient of mobile phase B for sample separation was: 10% to 40%, 0–5 min; 40% to 50%, 5–10 min; 50% to 100%, 10–18 min; and 100%, 18-22 min. The column equilibration was performed with 10% mobile phase B for 5 min after every sample measurement.

Peak extraction, annotation, identification, and lipid quantification were processed in Compound Discoverer v. 2.1 (Thermo Fischer Scientific, Waltham, MA, USA) as described previously [35]. When lipid isomers in the same lipid class and with the same carbon length and number of double bonds were identified, small letters were appended to their names so they could be sorted in order of retention time. The quantified peak areas of each lipid were normalized to that of the internal standard for the same or related lipid class, as seen in Table 2. DG, TG, and ChE were excluded from the subsequent quantitative analyses, because the relative SD of the peak area, of the corresponding internal standards in the extracted samples, was >30%. The normalized areas were divided by the protein content (μg) used for lipid extraction and regarded as the absolute lipid quantity per protein. To calculate the mol% of the lipids in the total phosphoglycerolipids and sphingolipids, the quantity of each lipid was normalized to the sum of all phosphoglycerolipids and sphingolipids per sample. The absolute lipid quantities in the test and validation samples are shown in Appendix A, respectively. The mol% of lipids in the validation sample are shown in Appendix A.

### 4.9. Statistical Analysis

The lipid profiles of the isolated fractions were examined by principal component analysis (PCA) in SIMCA+P14 (Umetrics, Umea, Sweden). Differences in the lipid class composition among the EVs, HDL, and LDL/VLDL were examined by one-way ANOVA followed by Tukey’s multiple comparison test in GraphPad Prism v. 8 (GraphPad Software, San Diego, CA, USA). For the heatmap analysis of the unique differences in individual lipids between the EVs and the lipoproteins, a very stringent criterion was applied (*p* < 1.89 × 10^−4^ = 0.05/264; Student’s *t*-test with a Bonferroni correction).

## Figures and Tables

**Figure 1 metabolites-09-00259-f001:**
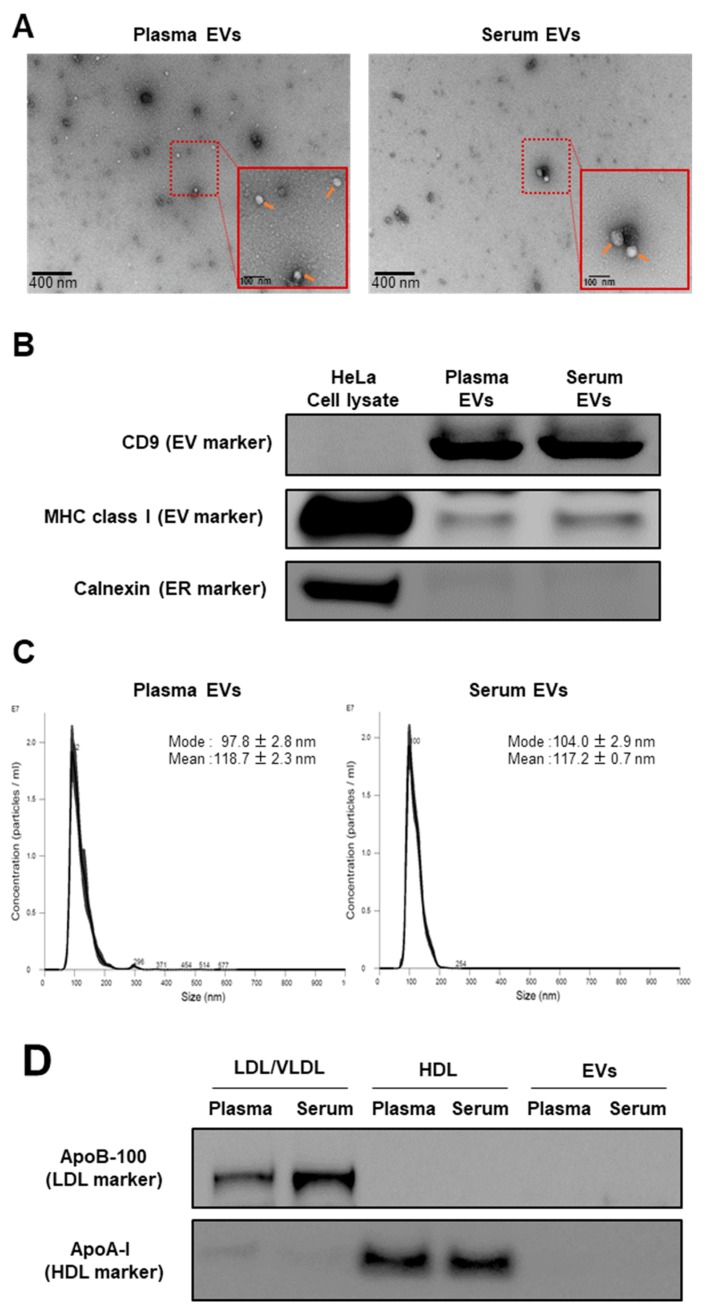
Validation of EV, high-density lipoprotein (HDL), and low-density lipoprotein (LDL)/very low-density lipoprotein (VLDL) extraction methods. (**A**) Identities of EVs isolated from pooled human plasma and serum were determined by TEM. Representative images are shown. Orange arrows indicate the isolated EVs. (**B**) Expression levels of the EV markers (CD9 and MHC class I) and endoplasmic reticulum (ER) marker (calnexin) in the isolated EV samples examined by Western blot. HeLa cell lysate was using as a positive control. (**C**) Particle size distribution of the isolated EVs assessed by the NTA. Measurements were performed 10 times. The mean and mode ± SE of the calculated particle sizes are shown in the graph. (**D**) Expression levels of apoB-100 (LDL/VLDL marker) and apoA-I (HDL marker) in the isolated LDL/VLDL, HDL, and EV fractions, examined by Western blot.

**Figure 2 metabolites-09-00259-f002:**
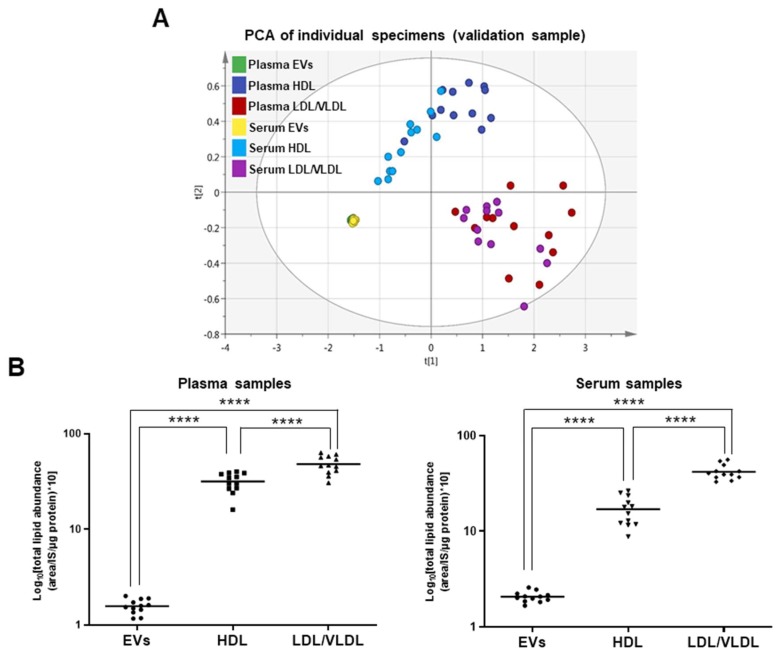
Comparison of lipid abundance in EV, HDL, and LDL/VLDL fractions in validation sample. (**A**) Lipid profiles of EVs, HDL, and LDL/VLDL obtained from human plasma or serum (*n* = 12 per group) analyzed by PCA. Analysis was performed using absolute lipid abundance (area IS^−1^ μg^−1^ protein). Goodness-of-fit parameters R2X and Q2 were 0.911 and 0.892, respectively. (**B**) Differences in absolute lipid abundance per microgram protein among EVs, HDL, and LDL/VLDL isolated from plasma or serum (*n* = 12 per group) examined by one-way ANOVA followed by Tukey’s test. Solid line in each group indicates mean total lipid abundance. **** *p* < 0.0001.

**Figure 3 metabolites-09-00259-f003:**
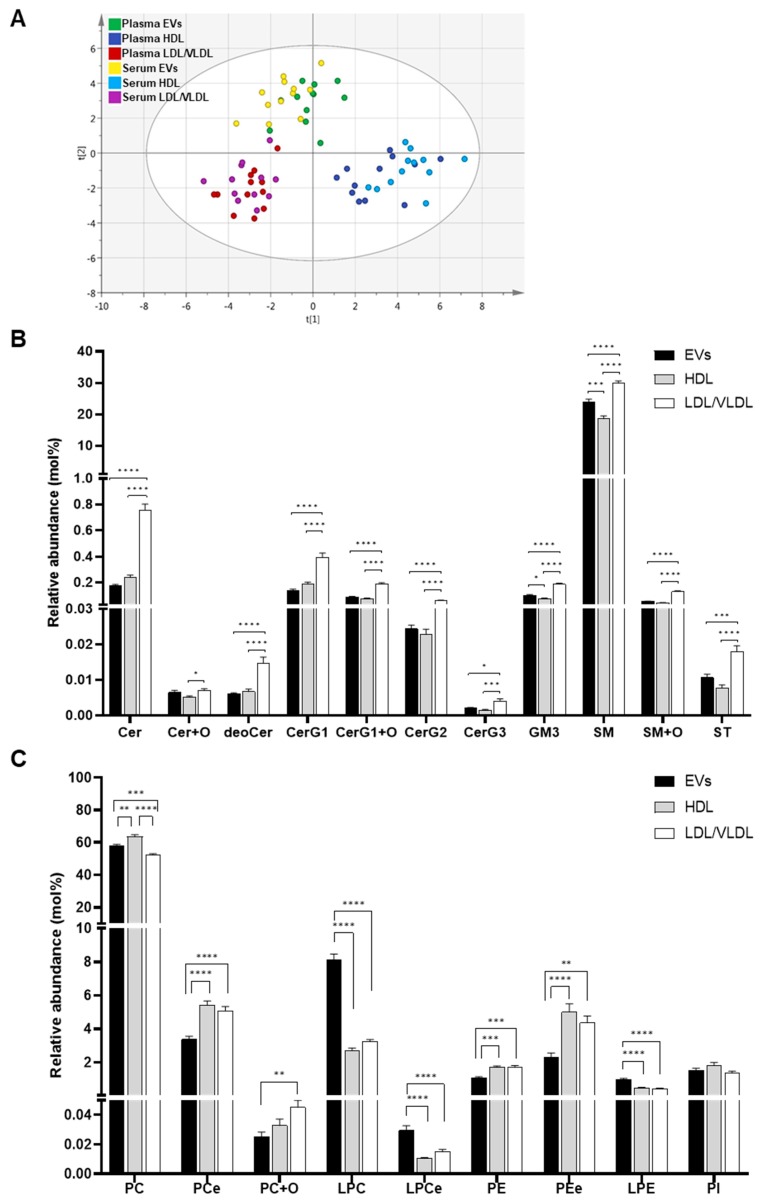
Comparison of relative concentration (mol%) of lipid classes in EV, HDL and LDL/VLDL fractions. (**A**) Lipid profiles of EVs, HDL, and LDL/VLDL obtained from human plasma or serum (n = 12 per group) analyzed by PCA. Analysis was performed using relative concentration (mol%). The goodness of fit parameters R2X and Q2 were 0.537 and 0.422, respectively. Relative lipid concentration (mol%) of sphingolipids (**B**) and glycerophospholipids (**C**). Relative concentration of each lipid class in EVs (black bars), HDL (gray bars), and LDL/VLDL (white bars) isolated from plasma (n = 12 per group). Bar graph plotted using average mol% values of each lipid class. Error bars indicate standard errors. Statistical differences among the groups analyzed by one-way ANOVA followed by Tukey’s test. * *p* < 0.05, ** *p* < 0.01, *** *p* < 0.001, **** *p* < 0.0001.

**Figure 4 metabolites-09-00259-f004:**
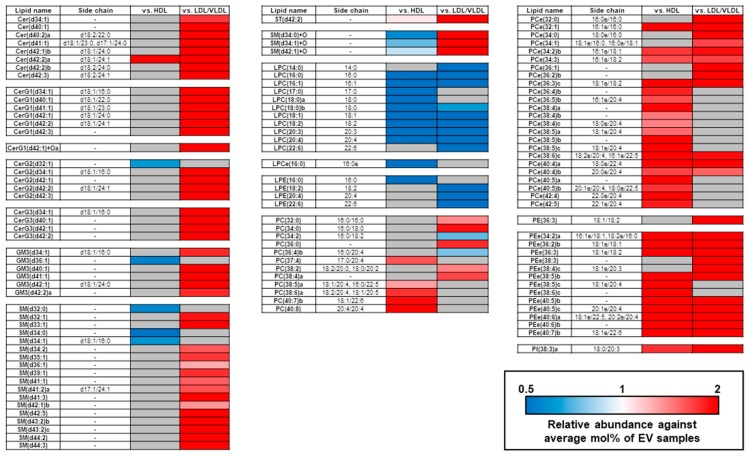
Differentially expressed lipid molecules in plasma EV, HDL, and LDL/VLDL fractions. Heatmap of relative lipid molecule concentrations in plasma EVs, HDL, and LDL/VLDL (n = 12 per group). Average mol% of each lipid molecule in plasma EVs was set to 1. Only lipid molecules showing statistical significance (*p* < 1.89 × 10^−4^ = 0.05/264; Bonferroni correction; Student’s *t*-test) between EVs and each lipoprotein, were assigned to build the heatmap.

**Figure 5 metabolites-09-00259-f005:**
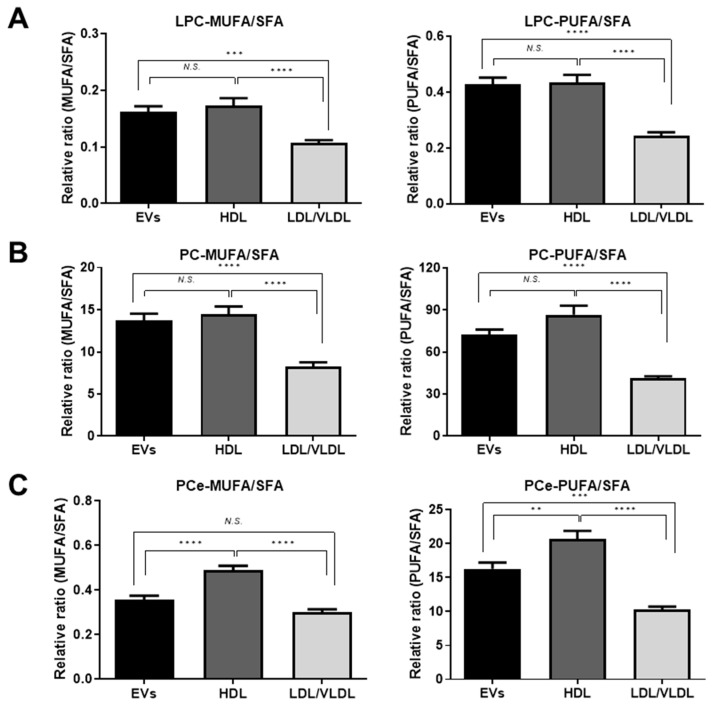
Differential composition of fatty acyl chains of PC-related lipid classes in plasma EVs, HDL and LDL/VLDL. Comparisons of monounsaturated fatty acid (MUFA):saturated fatty acid (SFA) ratio and polyunsaturated fatty acids (PUFA):SFA ratio in LPC (A), PC (B), and PCe (C) classes of plasma EVs, HDL, and LDL/VLDL (n = 12 per group). During analysis, PC species containing only one or >1 unsaturated bonds were classified as MUFA or PUFA, respectively. Those composed of only saturated fatty acids were classified as SFA. Relative ratios (MUFA:SFA and PUFA:SFA) were calculated for each lipid class. Statistical differences in MUFA:SFA and PUFA:SFA ratios among plasma EVs, HDL, and LDL/VLDL were examined by one-way ANOVA followed by Tukey’s test. ** *p* < 0.01; *** *p* < 0.001; **** *p* < 0.0001; *N.S.*; not significant.

**Table 1 metabolites-09-00259-t001:** Comparison of methodological differences in mass spectrometry-based extracellular vesicle (EV) lipidomics for human plasma and serum.

Materials and Methods	Our Method	Chen et al. [24]	Tao et al. [19]
Samples types	Human plasma and serum	Human serum	Human plasma
EV isolation	PureExo Exosome isolation kit	sequential ultracentrifugation	ExoQuick exosome precipitation kit
Lipoprotein contamination	Negligible	NE	NE
Lipid extraction	SPE	LLE	LLE
Lipidomics platform	LC-MS	Shotgun MS	LC-MS

NE, not examined; SPE, solid phase extraction; LLE, liquid-liquid extraction.

**Table 2 metabolites-09-00259-t002:** Lipids detected in fractionated samples.

Lipid Type	Class	Abbreviation	Internal Standards	# Lipids	
				Test Sample	Validation Sample
Glycerophospholipid	Lysophosphatidylcholine	LPC	PC(12:0/12:0)	14	11
	Ether-type lysophosphatidylcholine	LPCe	PC(12:0/12:0)	3	2
	Lysophosphatidylethanolamine	LPE	PE(12:0/12:0)	8	7
	Lysophosphatidylinositol	LPI	PI(12:0/13:0)	3	0
	Phosphatidylcholine	PC	PC(12:0/12:0)	55	51
	Ether-type phosphatidylcholine	PCe	PC(12:0/12:0)	52	43
	Oxidized phosphatidylcholine	PC+O	PC(12:0/12:0)	3	2
	Phosphatidylethanolamine	PE	PE(12:0/12:0)	17	14
	Ether-type phosphatidylethanolamine	Pee	PE(12:0/12:0)	34	26
	Phosphatidylinositol	PI	PI(15:0/18:1-d7)	17	12
Sphingolipid	Sphingomyelin	SM	SM(d18:1/12:0)	42	43
	Oxidized sphingomyelin	SM+O	SM(d18:1/12:0)	7	3
	Ceramide	Cer	Cer(d18:1/12:0)	17	15
	Oxidized ceramide	Cer+O	Cer(d18:1/12:0)	1	1
	Deoxyceramid	deoCer	Cer(d18:1/12:0)	1	1
	Glycosylceramide	CerG1	CerG1(d18:1/12:0)	9	8
	Oxidized glycosylceramide	CerG1+O	CerG1(d18:1/12:0)	3	2
	Diglycosylceramide	CerG2	CerG1(d18:1/12:0)	7	7
	Triglycosylceramide	CerG3	CerG1(d18:1/12:0)	5	4
	Ganglioside	GM3	CerG1(d18:1/12:0)	8	9
	Sulfatide	ST	ST(d18:1/12:0)	5	3
	Total		-	311	264

**Table 3 metabolites-09-00259-t003:** Sample data for healthy human subjects in this study.

Variable	Information
Sample type	plasma or serum
Gender	Male
Number of samples	12
Age	29 (26–33)
Body weight (kg)	72.1 (54.0–86.6)
Body height (cm)	181.6 (165.1–190.5)
BMI (kg m^−2^)	21.5 (17.8–26.6)

Parentheses indicate range (minimum–maximum) of each variable.

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
