# Peer review of "Lipid Profile Characterization and Lipoprotein Comparison of Extracellular Vesicles from Human Plasma and Serum"

_metabolites, 2019, doi:10.3390/metabo9110259_

Round 1

Reviewer 1 Report

Manuscript ID: molecules-623004

MANUSCRIPT SUMMARY

This study is a methodological description of lipid profiling (glycerophospholipids and sphingolipids) for extracellular vesicles (EVs) and lipoproteins in serum and plasma. A number of lipids were detected (~300) for test and validation samples. Overall the manuscript is well written and Figure 1 gives a nice summary of validation of the fractions.

This manuscript seems to make a nice reference for researchers specifically interested in lipids of EVs, but is very specific due to it being a methodology paper and lack of biological design limits its impact.

The novelty of the study is a little difficult to determine, there are statements that read as though this is the first lipid profiling study done on EVs, which is not the case and these should be correctly accordingly (see specific comments).  Are the authors stating that all previous studies on EV lipid profiling were contaminated by lipoproteins (line 222)? A table comparing this method to previous studies is lacking and should be included.

SPECIFIC COMMENTS

Abstract lines 12-13, “The aim of this study was to establish a lipidomics platform for human plasma and serum EVs, to characterize their lipid profiles, as this has not previously been done…” The statement is misleading, as there have been lipid-profiling studies on both lipoproteins and EVs. The comparison of EVs and lipoproteins is the novel aspect but the statement in the abstract does not read this way.

Line 46, change to “may” serve as a diagnostic biomarker…

Reviewer 2 Report

This comparative lipidomic characterization of liproteins and extracellular vesicles is a valuable resource for future research. The analytics appears to be well done. A major limitation is the isolation of lipoproteins and extracellular vesicles. The authors describe in the text only, that the isolated extracellular vesicles do not show any expression of apoA-I or apoB. This needs to be documented by a figure. Immunoblots of apoB and apoA-I should be added to figure 1B. The authors do not provide any information on the contamination of lipoproteins with extracellular vesicles. They should provide immunoblots of CD9, MHC class 1 and Calnexin in figure 1D. Since the authors isolated lipoproteins by the use of ultracentrifugation only, one must expect contaminations of lipoproteins with exosomes and microvesicular bodies which have a similar density. Because of the difference in size, these contaminations can be avoided by the combination of ultracentrifugation and gelfiltration. This reviewer assumes that the data on lipoproteins  reflect a mixture of lipoproteins  and  extracellular vesicles. If so,  differences may be  even more pronounced for some lipids, e.g.  LPCs, if lipoproteins  are separated entirely from the  vesicle fraction. 

Reviewer 3 Report

General notes;

The manuscript by Sun et al. presents an exploratory analysis characterising the lipidomic profile of extracellular vesicles and comparing them to lipoprotein lipidome. In general, the manuscript is well written and is of relative interest. The lipidomics is done quite well and the annotations appear quite appropriate.

Some general comments

While the exclusion of DG/TG/ChE was due to high variation in the samples, it is nonetheless important to report these species. While Western blot was provided to highlight non-contamination of lipoprotein particles, a sensitive marker of contamination would be the presence of ChE in the samples as they should be relatively low in EVs compared to LDL. A better description of the methodology would help the readers, so that they don’t need to chase down other papers. A brief description of the lipidomics instrumentation used, approaches and the separation technique would suffice. The conditions used for the lipidomics (reconstitution in methanol) with the auto sampler set at 4.5oc is likely the reason for high variation within the DG/TG/ChE. The addition of some non-polar solvent to the final reconstitution step along with higher auto sampler temperatures (~12 degrees) would stop precipitation of the non-polar lipids over time as it sits in the auto sampler.

Round 2

Reviewer 1 Report

The manuscript has been improved and should be of interest to researchers in the field.

Reviewer 2 Report

none